# Cardiovascular Disease Complicating COVID-19 in the Elderly

**DOI:** 10.3390/medicina57080833

**Published:** 2021-08-17

**Authors:** Christopher Dayaramani, Joshua De Leon, Allison B. Reiss

**Affiliations:** Department of Medicine and Biomedical Research Institute, NYU Long Island School of Medicine, Mineola, NY 11501, USA; cdayaramani@gmail.com (C.D.); Joshua.DeLeon@NYULangone.org (J.D.L.)

**Keywords:** COVID-19, atherosclerosis, cardiovascular disease, hypertension, inflammation, cytokines

## Abstract

SARS-CoV-2, a single-stranded RNA coronavirus, causes an illness known as coronavirus disease 2019 (COVID-19). The highly transmissible virus gains entry into human cells primarily by the binding of its spike protein to the angiotensin-converting enzyme 2 receptor, which is expressed not only in lung tissue but also in cardiac myocytes and the vascular endothelium. Cardiovascular complications are frequent in patients with COVID-19 and may be a result of viral-associated systemic and cardiac inflammation or may arise from a virus-induced hypercoagulable state. This prothrombotic state is marked by endothelial dysfunction and platelet activation in both macrovasculature and microvasculature. In patients with subclinical atherosclerosis, COVID-19 may incite atherosclerotic plaque disruption and coronary thrombosis. Hypertension and obesity are common comorbidities in COVID-19 patients that may significantly raise the risk of mortality. Sedentary behaviors, poor diet, and increased use of tobacco and alcohol, associated with prolonged stay-at-home restrictions, may promote thrombosis, while depressed mood due to social isolation can exacerbate poor self-care. Telehealth interventions via smartphone applications and other technologies that document nutrition and offer exercise programs and social connections can be used to mitigate some of the potential damage to heart health.

## 1. Introduction

Cardiovascular disease (CVD) is the leading cause of death worldwide [1,2]. Its incidence increases sharply with age, and the elderly bear a disproportionate burden of CVD morbidity and mortality [3,4,5]. Coronavirus disease 2019 (COVID-19), caused by severe acute respiratory distress syndrome coronavirus 2 (SARS-CoV-2), has also caused significant mortality, specifically amongst the elderly, who are the most likely patient population to be hospitalized and die from the infection [6,7,8]. 

Factors that likely contribute to a complicated course and higher death rate in COVID-19 patients with underlying CVD are the hypercoagulable state that can result from COVID-19 infection, as well as polypharmacy (an indicator of comorbidities) [9,10,11,12]. Furthermore, the pandemic can disrupt lifestyle, leading to poorer diet and inactivity [13]. Another obstacle for older CVD patients is the avoidance of medical care from fear of contracting COVID-19. Hypertension, diabetes, and obesity, which often accompany CVD, are themselves established risk factors for severe COVID-19 that require careful management [14]. There is also a higher prevalence of cancer within the elderly population [15]. In addition to being immunocompromised from the cancer itself, cancer patients are frequently treated with immunosuppressants and cardiotoxic chemotherapies, making them especially susceptible to both viral illnesses and secondary cardiovascular complications. A particularly high risk of poor outcome is seen in those who have undergone recent bone marrow or stem cell transplantation and those exposed to poly ADP ribose polymerase (PARP) inhibitors [16,17]. The aforementioned conditions are more prevalent in those over age 65 and extremely common in those over age 85. Low vitamin D levels, common in the obese state, may add further risk [18].

Excess production of cytokines and cytokine storms are central to many of the sequelae of COVID-19, including damage to the cardiovascular system via pathways that involve direct cardiotoxicity and through inflammation-induced myocarditis and pericarditis [19,20]. The effects of cytokines, particularly of interleukin (IL)-6, will be discussed in the sections to follow.

Since persons with CVD are susceptible to poor COVID-19 outcomes, targeted treatment and removal of barriers to care are crucial for this population [21]. These may include the use of telemedicine, adjusting antihypertensive regimens, and online activity tracking. In this review, focused on persons over age 65, we explore the interactions between CVD and COVID-19 that lead to increased vulnerability to serious illness and death. We then describe strategies to mitigate these risks.

## 2. Hypertension, the Immune System, and COVID-19

About 20% of COVID-19 patients are hypertensive, but some older COVID-19 populations have high blood pressure (HBP) rates exceeding 50%, consistent with the prevalence of hypertension in persons of advancing age (Table 1) [22,23,24]. The status of hypertension as an independent risk factor for morbidity and mortality in COVID-19 is unclear. Data from China indicate a link between HBP and more severe COVID-19 infection [25]. A meta-analysis of 9 studies from China examining COVID-19 comorbidities found a significant correlation between COVID-19 severity and history of hypertension [26,27]. Based on pooled data from another meta-analysis from China, hypertension conferred a 2.5-fold increase in risk of severe COVID-19 illness and a 2.4-fold increase in mortality risk [28]. This relationship between hypertension and COVID-19 was also found, but to a lesser degree, in a community cohort study of persons aged 65 and older in England, where hypertension was found in 59.6% of 502 hospitalized COVID-19 patients and associated with a 1.38-fold greater risk for hospitalization but was not significantly associated with mortality [29]. A cross-sectional study of over 210,000 COVID-19 patients from Mexico found that a number of comorbidities, including hypertension, increased mortality [30]. Of these cases, over 20% were hypertensive, and hypertension increased the risk of death (adjusted odds ratio 1.24). Among older hospitalized COVID-19 patients in Italy, a retrospective study of 87 persons with an average age of 72 years found that over 50% were hypertensive but that hypertension did not predict mortality [31]. A case series of 5700 adults, median age 63 years, hospitalized in the New York City area for COVID-19, found that 56.6% of the patients had HBP, but the report did not address differences in outcome based on comorbidities or medications [18]. A study of 184 COVID-19 patients with a mean age of 64.7 years, admitted to a New York City hospital, found a positive correlation between hypertension and mortality [32]. In the United States, a multicenter cohort study of 2215 patients hospitalized in ICUs for COVID-19 (59.7% with hypertension, mean age 60.5 years) found that HBP did not increase the risk of death [33]. 

Numerous confounding variables make it difficult to link hypertension directly to COVID-19 severity [38]. Bajgain et al. recently posited that comorbidities such as hypertension may increase disease severity without escalating mortality [34]. Hypertension may increase hospital readmission rates after discharge for COVID-19 patients, with re-hospitalization most often due to respiratory distress or thrombotic episodes [35].

Although a contribution to COVID-19 mortality from hypertension is unconfirmed, increased susceptibility to severe illness may result from a chronic hypertension-induced inflammatory state causing immune dysregulation [39]. Basal levels of proinflammatory cytokines are increased in patients with hypertension, where ILs-6, -8, -18, transforming growth factor (TGF)-β, and tumor necrosis factor (TNF)-α are in disequilibrium with anti-inflammatory cytokines (ILs-4 and -10) [40].

Hypertension affects CD8+ T-cells, essential components of cell-mediated immunity that protect against acute and persistent viral infections [41]. Hypertension stimulates CD8+ overproduction of cytokines, most notably, interferon (IFN)-γ and TNF-α [42,43]. These cytokines can adversely affect renal epithelium and raise blood pressure by increasing oxidative stress and renal sodium reabsorption [44,45]. Lymphopenia, due to reductions in both CD4+ and CD8+ T-cells, is well-documented in severe COVID-19 and may further disable antiviral defenses [36]. Pan and colleagues found that hypertensive COVID-19 patients have lower CD8+ counts, which may compromise the defense against the damaging flood of cytokines, characteristic of COVID-19 infection [37,38]. 

Another key potential link between COVID-19 and hypertension is IL-6, a pleiotropic cytokine found at high levels in cytokine storms, which are responsible for stimulating the production of acute-phase proteins [46,47]. Plasma IL-6 levels strongly correlate with hypertension in humans [48]. In the initial phase of inflammation, IL-6, together with TGF-β, is necessary for the development of Th17 cells, which help induce a state of neutrophilic inflammation [49]. Chronic sustained IL-6 elevation can result in cardiac fibrosis and left ventricular hypertrophy [50,51,52]. High IL-6 levels are associated with an increased risk of death from COVID-19 infection [53,54,55]. Whether the IL-6 level rather than blood pressure is the independent variable determining COVID-19 severity risk is unresolved [56]. 

The possibility that IL-6 worsens COVID-19 outcomes has led to anti-IL-6 therapy trials with tocilizumab and sarilumab, but the results have been mixed [57,58,59,60]. A recent meta-analysis showed a significant reduction in all-cause mortality with a combination of tocilizumab and dexamethasone in patients of varying ages, with the median age of approximately late 50s to early 60s [61]. In a randomized study, both tocilizumab and sarilumab were effective in improving the survival of severely ill ICU patients (mean age of about 60 years) with COVID-19 [62]. It is possible that IL-6 either suppresses or increases viral replication and persistence, depending on multiple factors [63,64,65]. 

## 3. Angiotensin-Converting Enzyme Inhibitors/Angiotensin Receptor Blockers (ARB) in COVID-19

The renin–angiotensin system (RAS) plays a key role in blood pressure regulation and sodium balance [66]. In the classical RAS pathway, renin cleaves the proenzyme angiotensinogen to form angiotensin (Ang)I. AngI is further cleaved by the angiotensin-converting enzyme (ACE) to yield AngII, which then acts through angiotensin receptor types 1 and 2 (AT1, AT2). The AngII/AT1 receptor axis mediates vasoconstriction, while the effects of AT2 are often antagonistic to AT1 (Figure 1). Vasoconstriction of the pulmonary vasculature can lead to right ventricular hypertrophy [67,68]. AngII also increases blood pressure indirectly by stimulating aldosterone production, which increases sodium and water reabsorption [69].

ACE2, homologous to ACE, cleavages AngII via an alternative pathway to form Angiotensin 1–7. Angiotensin 1–7 has vasodilatory effects and protect against acute respiratory distress syndrome (ARDS) [70]. ACE2 is constitutively expressed in the lungs, kidney, gastrointestinal tract, and endothelium. The COVID-19 particle enters host cells via the full-length membrane-bound ACE2, which has an extracellular domain that acts as a receptor for the SARS-COV-2 spike protein [71]. The spike transmembrane glycoprotein is the target of many mRNA vaccines against COVID-19 [72]. 

Cleavage of the viral S-spike protein by host proprotein convertases generates S1 and S2 segments. The S1 (globular) subunit contains the receptor-binding domain that allows the attachment of COVID-19 to the ACE2 receptor. Once attached, COVID-19 fuses with the host cell membrane via the S2 subunit, enters the cell, and subsequently replicates in the cytoplasm [73,74]. Hypertension may also increase ACE2 enzyme concentration, which initially sparked concern that hypertension might increase viral entry into cells, amplifying the severity of infection [75]. Long-term use of ACE inhibitors (ACEIs) or ARBs can upregulate ACE enzymes. ACEIs are not known to increase pulmonary or serum ACE2 levels but do increase ACE2 in the brain, heart, and urine [76,77]. Overall, there exists no definitive evidence that ACEIs or ARBs increase COVID-19 risk [78,79,80,81,82,83]. In fact, a preponderance of evidence indicates ACEIs and ARBs are not harmful to COVID-19 patients and may even provide some benefit [84,85,86]. In a cohort from France of over 2 million hypertensive patients (mean age of 63 years), the outcome in patients with COVID-19 on ACEIs and ARBs was compared to those on calcium channel blockers. The ACEI and ARB groups showed a lower risk of COVID-19 hospitalization and a lower risk of intubation/death [87]. 

A retrospective single-center study from Long Island, New York, found that 614 of over 6000 COVID-19 patients were hypertensive and required hospitalization. These patients were categorized as taking and not taking ACEIs/ARBs prior to admission. Those who had been on ACEIs/ARBs and discontinued these medications during their hospital stay were twice as likely to be admitted to the ICU as those who continued their established treatment. The authors speculate that, although the medications may cause ACE2 receptor upregulation, discontinuing hypertensive medication can worsen illness because the benefits of controlling blood pressure outweigh other factors and, possibly, ACEIs/ARBs may mitigate the effects of cytokine storms [88]. This does not, however, indicate whether a change in medication prior to contraction of illness or onset of symptoms would confer benefit. Genetic variations in RAS genes and their expression profiles may also influence how pharmacotherapies affect COVID-19 outcomes; this is an area of active study [89,90]. 

The current recommendation is that patients should not change their regimen, as directed by the European and American Societies of Cardiology as well as the AHA [75,81,91,92,93]. 

## 4. Pulmonary Arterial Hypertension and Right Ventricular Hypertrophy

Pulmonary arterial hypertension (PAH), defined as a mean pressure above 25 mmHg in the pulmonary circulation at rest (normal = 24/12), confers susceptibility to vascular remodeling, fibrosis, and pulmonary edema [94]. Severe and sustained PAH can manifest as pulmonary edema and right-sided hypertrophy and/or heart failure. The incidence of COVID-19 in persons with PAH is comparable to that in the general population, and based on their documented propensity to worse outcomes during hospitalization in general, they may have greater mortality during COVID-19-related hospitalization [95,96]. In 24 COVID-19 patients with PAH, Pagnesi et al. determined that PAH specifically, and not right ventricular hypertrophy, was associated with more severe COVID-19 illness and worse clinical outcomes [97]. 

The RAS is of clinical importance in both PAH and COVID-19. Within the RAS, ACE2 may offer vasoprotective effects. ACE2 supplements are being considered as a treatment for several lung diseases [98]. Such supplements have ameliorating effects in PAH via increased Angiotensin 1–7 and nitric oxide bioavailability. Inhaled nitric oxide was shown to be effective in the 2003 SARS epidemic, causing the reversal of PAH and decreased lengths of ventilator support. Clinical trials are evaluating this therapy in COVID-19 [99,100,101,102,103,104,105,106,107,108,109,110,111]. PAH is characterized by increased levels of the vasoconstrictor endothelin-1, and, therefore, along with increasing nitric oxide, the antagonism of the endothelin-1 receptor is effective PAH therapy that may be beneficial in COVID-19 infection [102]. Endothelin-1 may decrease ACE2, augmenting vasoconstriction via higher levels of AngII from lung epithelial cells [103,104,105]. Endothelin-1 may also promote endothelial dysfunction, platelet activation, and IL-6 secretion, all detrimental in the setting of COVID-19 [106,107].

Overall, COVID-19 is known to induce microthrombosis and hemorrhage in the alveoli and lung interstitium, and this may be exacerbated by pre-existing hypertension or increased vascular stress brought on by COVID-19-associated PAH. [108]. 

## 5. Coagulopathy in COVID-19: Mechanisms, Manifestations, and Treatment

A state of hypercoagulability frequently accompanies COVID-19, especially in severe disease [109,110]. Coagulopathy leaves patients vulnerable to thrombotic complications, including venous thromboembolism, pulmonary embolism, and disseminated intravascular coagulation [111,112,113,114,115].

Pro-inflammatory mediators produced during COVID-19 infection cause the release of tissue factor, an initiator of blood coagulation, from mononuclear cells [116,117]. IL-6, a key mediator elevated in the COVID-19 setting, can raise tissue factor levels and may also stimulate platelet production in bone marrow and lungs [118,119,120]. The panoply of inflammatory factors also activates endothelial cells, increasing their expression of adhesion molecules and leading to the release of the von Willebrand factor, thus promoting a procoagulant endothelial phenotype, excessive activity in the coagulation cascade, and multiple thrombotic complications. 

Coagulation abnormalities are detected in laboratory tests as increased serum concentrations of the procoagulants fibrinogen and D-dimers as well as decreased antithrombin and prolonged prothrombin time (Table 2) [121,122].

As a result of vascular injury, the propeptide fibrinogen is cleaved to fibrin, and high circulating fibrin levels are common in the early phase of COVID-19 infection. Either hyper- or hypofibrinolysis can occur in the setting of COVID-19, with hypofibrinolysis causing susceptibility to thrombus formation and hyperfibrinolysis causing susceptibility to bleeding [126]. The hypofibrinolytic state may be attributed to the elevated production of plasminogen activator inhibitor 1 (PAI-1) by epithelial and endothelial cells in the inflamed lung [127]. D-dimers, produced during the degradation of crosslinked fibrin, are below 0.5 μg/mL under normal physiologic conditions. An increase in fibrinogen and D-dimers is associated with the risk of microthrombus formation in COVID-19 patients and subsequent emboli and/or organ failure [128,129]. Fibrinogen levels were, on average, higher in patients who developed severe versus less severe illness (5.16 vs. 4.51 g/L) [130]. D-dimer levels over 1 μg/mL can identify patients with poorer prognoses early in the course of disease and may signal the need for admission to critical care. Elevated D-dimer appears to be an independent risk factor for death [130,131]. In evaluating D-dimer levels, the implementation of age adjustment instead of a fixed cutoff may increase the accuracy of clinical assessment [132].

Sepsis-induced coagulopathy due to COVID-19 infection can lead to thrombotic stroke and myocardial infarction (MI) [133,134,135,136]. Losartan, previously mentioned for its ability to normalize levels of ACE2, is believed to be protective against strokes, offering another reason for its use in place of ACEIs [137].

Addressing the hypercoagulability risk at all levels of COVID-19 severity and in different age groups is challenging. Hypercoagulability may worsen in the setting of this pandemic via decreased activity, decreased exercise, and less movement in general under quarantine restrictions. Particularly affected are the elderly, with more limited movement capability due to age and co-morbidities. Venous stasis that accompanies inactivity, combined with hypercoagulability, sets the stage for two of three predisposing factors described in Virchow’s triad for vascular thrombosis (blood flow alterations, endothelial injury, and hypercoagulability) [138]. Hypertension predisposes to endothelial injury, the last remaining factor in Virchow’s triad. As discussed above, ACEIs should be used cautiously in COVID-19 patients. Anti-inflammatory drugs, cytokine inhibitors, and statins may be considered to protect the endothelium while simultaneously working against viral replication. Thromboprophylaxis with low molecular weight heparin could decrease D-dimer levels, and heparin can decrease fibrosis in those suffering from COVID-19-induced ARDS [130]. At this time, there are different approaches to coagulopathy in the field, with the evaluation of efficacy still ongoing. We await the results of major clinical trials regarding the dosage, class, and timeline of the use of anticoagulation therapy [139].

## 6. Myocardial Injury: Subclinical Atherosclerosis and Acute Coronary Syndrome 

COVID-19 cardiac manifestations may include myocarditis, cardiac arrhythmias, and new or worsening heart failure, which may be particularly damaging to patients with a history of CVD [140,141]. The mechanisms underlying cardiac injury may be multifactorial (Figure 2). Inflammation and thrombosis are known culprits [142]. Infection increases overall metabolic demand and heart rate. This intensifies oxygen expenditure while shortening the filling time in the diastole and limiting coronary perfusion. Infection-mediated vasoconstriction and ventilation/perfusion mismatch negatively affecting blood oxygenation can exacerbate the oxygen deficit, leading to myocardial ischemia [143,144]. Internalized virus loads within cardiomyocytes may directly damage the heart. 

Immune-inflammatory-mediated injury to the heart from COVID-19 is more likely in severe cases and in those with HBP and can be monitored via the release of certain injury biomarkers, including cardiac-specific troponin and creatine kinase-MB [145,146,147,148]. Elevated lactate dehydrogenase (LDH) may be of cardiac origin but is nonspecific and can result from damage to other organs [149]. In COVID-19, LDH may be released from the lung since this is a key site for inflammatory processes [150]. LDH levels above 445 μg/mL on admission can be predictive of more severe COVID-19 [133,151,152,153]. An analysis of 353 COVID-19 patients, 79 (22.4%) of whom presented with myocardial injury, revealed more frequent elevations in LDH (mean level: 244 U/L (without MI) vs. 655 U/L (with MI) and creatinine (71 μmol/L (without MI) vs. 155 μmol/L (with MI) in the MI group during their hospitalization [154]. In addition to acute MI, differential diagnoses for increased serum cardiac biomarkers include stress-induced cardiomyopathy as well as myocarditis [155,156,157].

Evidence is accumulating that cardiac injury combined with COVID-19 infection, whether the myocardial injury is pre-existing or occurs during the infection, is associated with poorer outcomes [158,159,160]. A prospective, multicenter cohort study in Spain found that in patients with acute MI, COVID-19 is an independent risk factor for in-hospital mortality [161]. A small study of 77 COVID-19 patients who died in Wuhan, China, in early 2020 found that heart disease was present in 32%, and heart disease patients were more likely to be in the short-term survival group and to die within 14 days of COVID-19 onset [153].

COVID-19-associated coagulopathy can cause both large and small vessel embolic phenomena [123,162]. Postmortem histological evaluation has revealed signs of microvascular thrombosis in both skin and lungs [163]. Coronary arteries and coronary microvasculature may be affected by COVID-19 because this viral infection can directly activate the endothelium via inflammatory cytokines and indirectly by causing hypoxic conditions [124,164,165]. Further, the virus can directly infect endothelial cells, leading to an inflamed endothelium [166]. Coronary microvascular endothelial inflammation could lead to myocardial injury [125]. The potential for COVID-19 to compromise the patency of coronary bypass grafts has been suggested [167].

Subclinical atherosclerosis can impact the course of COVID-19. Coronary artery calcification (CAC), a specific imaging marker of coronary atherosclerosis that correlates with the plaque burden, can reveal previously undiagnosed CVD in COVID-19 patients [168]. In a cross-sectional study of 209 consecutively admitted COVID-19 patients without known CVD, assessed for CAC, CAC was detected in 106 (Table 2) (50.7%). Half of those positive patients required mechanical ventilation, extracorporeal membrane oxygenation, or died, whereas only 17.5% of patients negative for CAC had such poor outcomes [169]. A separate study by Nai Fovino et al. from Italy also found that high CAC as a surrogate for subclinical atherosclerosis was a marker for worse outcomes [170]. In this study, 75% of patients with high CAC either died or were admitted to the ICU, in contrast with only 20% of the group with lower CAC scores. Patients with a high score were also more likely to experience an MI.

C-reactive protein (CRP), an acute-phase reactant and marker of chronic low-grade systemic inflammation, has prognostic value as a predictor of cardiovascular risk. A number of studies have shown an association between elevated CRP and higher rates of COVID-19-related hospitalization and mortality (Table 2) [171,172]. While moderate CRP elevation in the 3–10 mg/L range is useful in cardiovascular risk prediction, CRP values in COVID-19 may exceed 40 mg/L, reflecting an acute inflammatory response unrelated to the heart [173,174]. Ferritin, critical for iron homeostasis, is also associated with a high risk of arterial hypertension and has utility in predicting the risk of ARDS in COVID-19 patients [175,176]. Ferritin is used to monitor systemic inflammation and may also be considered a marker of a hypercoagulable state [177]. Hypoxia is also a factor in elevating ferritin levels, and, thus, high ferritin may be a consequence of poor oxygenation [178]. Ferritin-targeted therapies are promising as a means of limiting the hypercoagulable state in the early phases of COVID-19 [179,180].

Patients with pre-existing CVD and atherosclerotic plaque containing inflammatory cells may be vulnerable to plaque activation upon exposure to cytokines such as IL-6.and IFN-γ [24,181,182,183]. Leukocyte infiltration and the activation of the endothelium contribute to the host synthesis of metalloproteinases and peptidases that break down collagen and weaken and destabilize plaque [184,185]. Furthermore, acute infection induces a prothrombotic, hypercoagulable state that can incite the formation of platelet thrombi [186]. 

## 7. Lifestyle

In response to the COVID-19 public health emergency, many governments imposed lockdown measures, stay-at-home orders, and quarantine. Curfews are in effect in many parts of the world, and social distancing is strongly recommended or mandated [187,188,189,190]. Quarantine refers to the separation of individuals or communities who may have been exposed to an infectious disease, while lockdown is a restriction in movement, with the closing of non-essential businesses and activities. 

Quarantine and lockdown are effective in preventing disease transmission but can increase sedentary behavior both, directly through impeded access to exercise facilities and outdoor activities and the shutdown of interactive sports as well as indirectly through an increased prevalence of anxiety, depression, and insomnia that drain energy [191,192,193,194,195]. Sedentary behavior and low physical activity are of particular concern in those with CVD as these behaviors are highly associated with increased mortality in both males and females [196,197,198]. Physical activity has numerous favorable effects on the cardiovascular system, while lack of physical activity has long been linked to atherosclerosis [199,200]. Exercise delays the loss of age-associated endothelium-dependent vasodilation and reduces systemic inflammation and oxidative stress [201,202,203,204]. It also improves cardiac function and the lipid profile and decreases the resting heart rate [205,206,207]. Moderate to vigorous physical activity, 60 min per day, 5–7 days per week, is recommended to combat the physical and mental toll of COVID-19 isolation. Outdoor activities may include walking or jogging; if the workout is indoor only, home-based stairclimbing, walking with step-counts, or an exercise bicycle are good options [208,209].

Another lifestyle factor of concern for these patients during the pandemic is diet. To compound the issue, quality of diet and level of physical activity are linked, and people who spend a lot of time in front of screens also tend to eat more processed foods [210,211]. Pandemic constraints can lead to increased consumption of less healthy foods [212,213]. Boredom, anxiety, hesitation to venture outside, and fear of food shortages can drive the purchase of more salty, sugar-laden foods with high trans-fat content and longer shelf life. Poor dietary choices may result in weight gain and worsen pre-existing diabetes and hypertension [214]. A survey conducted in Italy found that 48% of participants noted weight gain during quarantine [215].

A balanced, nutritious diet containing fresh fruits and vegetables is important for the maintenance of healthy body weight, serum lipid profiles, glucose metabolism, and blood pressure levels [216,217,218,219,220]. Grocery delivery services are often available and particularly useful to susceptible populations, such as those suffering from CVD and the elderly. Of note, an anti-inflammatory diet, high in fiber from legumes, vegetables, and whole grains, can promote the production of short-chain fatty acids that may offer protection via the inhibition of cytokine production. Overall, this diet could potentially reduce the risk of cardiovascular complications of COVID-19 by promoting healthy weight, improving glucose control, and regulating metabolism [221]. Specific nutraceuticals of benefit to the cardiovascular system in COVID-19 are not yet defined, but the area is under study [222].

Quarantine may introduce stressors into life such as the loss of childcare resources, unemployment, death of a loved one, or strained relationships. Increased stress may lead to substance abuse. A study conducted in Poland showed that alcohol consumption increased in 14.6% of survey respondents, and 45% of smokers acknowledged smoking more. Furthermore, 43% of respondents reported increased food consumption/snacking [223].

Lastly, quarantine and feelings of isolation can increase depression and anxiety [224]. Among 1593 participants, Lei et al. found the total prevalence of anxiety and depression to be approximately 8.3% and 14.6%, respectively, and the prevalence in the isolated group (12.9%, 22.4%) was significantly higher than that in the group that did not undergo quarantine (6.7%, 11.9%) [198]. The length of quarantine is also highly associated with the development of mental health issues, with persons under quarantine conditions for more than 10 days more likely to develop post-traumatic stress than those quarantined for less than 10 days [190]. Mental well-being can be impacted by fear of contracting the illness, fear of the financial burden associated with quarantine, restricted availability of food, lack of information regarding quarantine specifics, and excessive time spent on COVID-19 related news (>1 h) [225,226,227].

Quarantine can lead to increased suicide risk, particularly among the elderly, who are more likely to live alone and perceive themselves as a burden [228]. People with underlying CVD who are subjected to extended periods of quarantine may experience stress-related worsening of their cardiac status [229,230].

## 8. Delivering Care

Care delivery to patients with COVID-19 in office or hospital settings comes with the risk of transmission and spread to physicians, staff, and other patients. Telemedicine has been slowly gaining acceptance for the advantages it offers in access to care and cost-effectiveness. Now, the pandemic has accelerated its growth because it brings the key benefit of increased safety through the circumvention of direct physical contact [231]. This safety comes with a downside as fewer in-person visits to medical personnel can lead to decreased patient engagement [232]. The Technology Acceptance model states that the willingness of the patient to utilize technology depends on two key factors: perceived usefulness and perceived ease of use [233]. Therefore, barriers to use, such as lack of familiarity with technology, privacy concerns, or preference for in-person communication, may impact patient interaction and satisfaction with telemedicine [234,235].

Interestingly, in the case of hypertension, several studies have shown that virtual approaches are as effective or more effective than traditional treatments in achieving blood pressure goals [236,237,238]. ECG transmission systems may allow the detection of acute MI in real-time [239,240]. It is important to note that cardiac emergencies require in-person evaluation and treatment without delay.

## 9. Tailoring CVD Treatment and Vaccination in the Context of COVID-19

As of this writing, our understanding of the pathophysiologic mechanisms of Sars-Cov-2 infection is rapidly changing, and evidence thus far obtained continues to include potentially conflicting hypotheses [241,242]. As such, identifying targets for therapy remain elusive, and no definitive recommendations can be made. In addition, previously used therapies and some therapies under investigation may have adverse cardiovascular effects, while their clinical efficacy for combating COVID-19 is unconfirmed [243]. However, once specific cardiovascular complications have been identified, such as acute coronary syndromes, acute heart failure, or myocarditis, it is reasonable to employ standard treatments and therapies. Additionally, regardless of the exact mechanism(s) resulting in cardiovascular involvement, the severe inflammatory state associated with this disease pinpoints targeting this state as an important therapeutic goal. As demonstrated in their review, Samidurai and Das [244] note that the precise mechanisms linking CVD and worsened prognosis or higher mortality rate in COVID-19 patients remain unknown, but it is presumed that the pathophysiology of this virus’ method of infection, because it may directly involve the cardiovascular system, predicts high rates of cardiovascular involvement and higher morbidity and mortality in patients with pre-existing CVD.

As noted previously, it is generally accepted within the medical community that ACE inhibitors and ARBs should be continued as treatment in hypertensive patients who have been diagnosed with COVID-19 [245,246]. In those with diabetes, COVID-19 may increase blood glucose levels, and, therefore, strict monitoring and blood sugar control may be indicated [247,248].

In those with CVD, who may be vulnerable to severe effects of COVID-19, prevention of illness by vaccination is preferable to treatment after infection. At the time of this writing, mRNA vaccines and adenoviral vaccines targeting the spike protein are being administered widely [249]. The American College of Cardiology recommends that persons with CVD be prioritized for vaccination and, thus far, the vaccine is well-tolerated by these patients [250,251]. A small study of 86 mostly young subjects from Rome found that hypertension, central obesity, and smoking were each associated with lower antibody titers in individuals vaccinated with the Pfizer-BioNTech mRNA vaccine and suggested that individuals with these risk factors may benefit from a booster vaccine dose [252].

One caveat is the possible link between COVID-19 mRNA vaccination and myocarditis, but this is largely seen in younger vaccine recipients with a median age of about 26 years [253,254]. A major concern is the diminished effectiveness of vaccination in those who have undergone solid organ transplants, including heart transplants. These patients may fail to mount a robust immune response to vaccination, and this lack of response is more likely in older persons above age 60. [255,256]. A booster vaccine dose may improve protection against illness in these patients [257].

## 10. Conclusions

For the reasons discussed above, the elderly with preexisting CVD are vulnerable to multiple cardiovascular manifestations that can lead to adverse outcomes. Table 3 highlights the most pressing issues facing older persons with CVD during the COVID-19 pandemic. It is our hope that with the release of highly effective vaccines, prevention will render treatment needs less urgent in the future, but as the virus evolves, new variants may prolong the pandemic [258]. Identification of effective therapies and novel therapeutic targets remain a central focus for research and development [259]. Preventive strategies such as advanced technological tools, including artificial intelligence and machine learning, can accelerate the diagnosis/screening of patients for the virus [260]. Rapid COVID-19 testing and self-testing are becoming more accessible [261]. In addition, analysis of available literature and identification of potential therapeutic targets and other specific clinical features can help address this contagion and perhaps form the foundation for addressing future pandemics.

## Figures and Tables

**Figure 1 medicina-57-00833-f001:**
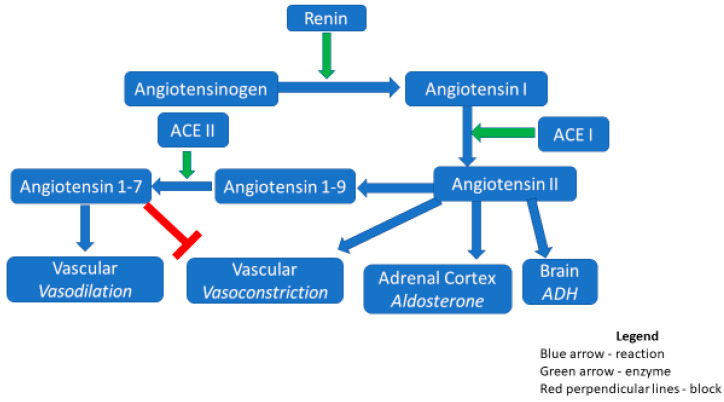
Renin–angiotensin-aldosterone system (RAS) and ACE inhibitors. The precursor molecule angiotensinogen is cleaved by renin to form angiotensin 1 in response to low sodium-chloride, as sensed by the macula densa. Angiotensin 1 is then further modified by the angiotensin-converting enzyme (ACE) to form angiotensin II. The ACE is the target of ACE inhibitors. Angiotensin II has direct effects on the vasculature, adrenal cortex, and brain, causing vasoconstriction and the secretion of aldosterone and the anti-diuretic hormone (ADH), respectively. An alternate pathway also exists, whereby angiotensin II is converted to angiotensins 1–9 and then angiotensins 1–7. A key enzyme in the formation of the latter is ACE2. Angiotensins 1–7, in contrast to angiotensin II, cause vasodilation and inhibit vasoconstriction.

**Figure 2 medicina-57-00833-f002:**
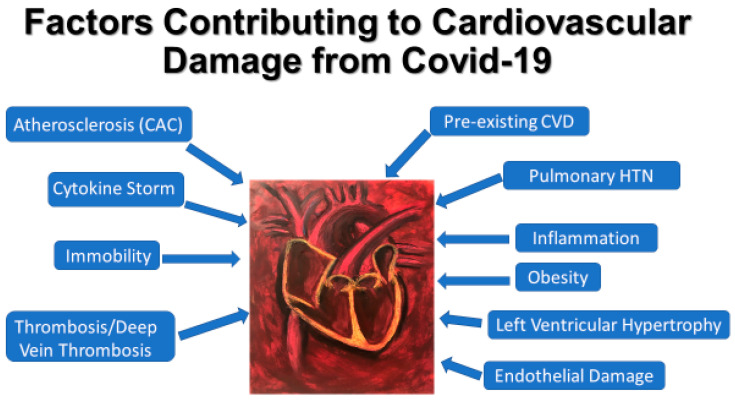
Factors contributing to cardiovascular damage in COVID-19. The white text in blue bubbles represents some of the pathologies that are associated with CVD in patients with COVID-19. Arrows are drawn from each factor to the central image of the heart and associated vasculature to emphasize their effect on the cardiovascular system specifically. CAC—Coronary Artery Calcium score; IL-6: interleukin-6; CVD: cardiovascular disease.

**Table 1 medicina-57-00833-t001:** Hypertension and COVID-19 studies, with age included.

By First Author	Reference	Age Range	Details
Baradaran et al.	[22]	Mean age 51 years; 95% confidence interval, 49–54 years	Meta-analysis
Richardson et al.	[23]	Median age 63 years; interquartile range, 52–75 years	5700 subjects, New York City area, USA
Ahrenfeldt et al.	[24]	Mean age 68 years; age range 50–104 years	73,274 subjects, European region
Wu et al.	[25]	Median age 51 years; interquartile range, 43–60 years	201 subjects, Wuhan China
Chen et al.	[26]	Unspecified	Systematic review, 1936 subjects
Liu et al.	[27]	Median age 52.4 years; age range 32.5–64.0 years. Severe group older than non-severe group (60.9 years, range 45.0–74.0 years)	Meta-analysis, 10,948 subjects, most from China
Lippi et al.	[28]	COVID-19 severity associated with hypertension observed only in subjects over age 60 years	Systematic review, 2893 subjects, China
Atkins et al.	[29]	Mean age 74.3 years (SD 4.5)	UK Biobank Cohort, 507 COVID-19 positive subjects, hospitalized
Hernández-Galdamez et al.	[30]	45.7 (SD 16.3)	Cross-sectional study, 211,003 subjects, Mexico
De Vito et al.	[31]	Median age 72 years; interquartile range, 62.5–83.5 years	Retrospective, single-center study, 87 subjects, Italy
Mani et al.	[32]	Mean age 64.72 years (SD 14.87)	184 subjects, New York City area, USA
Gupta et al.	[33]	Mean age (all) 60.5 years (SD 14.5). Mean age (died) 66.0 years (SD 13.3)	Multicenter cohort study, 2215 subjects, USA
Bajgain et al.	[34]	Median age 56 years; interquartile range, 48.25–67.4 years	Systematic review, 22,753 subjects
Atalla et al.	[35]	Median age 61 years; interquartile range, 49–74 years	339 patients, Rhode Island, USA
Zheng et al.	[36]	Median age 47.13 years; range, 11–84 years.	Cohort study, 68 patients, China
Pan et al.	[37]	Median age of hypertensive patients 69 years; interquartile range, 62–76 years	Single-center, retrospective study, 996 patients, China

SD: standard deviation.

**Table 2 medicina-57-00833-t002:** Early Markers Associated with Poor Outcomes in COVID-19 Patients.

Marker	Mean Level	Time of Measurement	Definition of Poor Outcome	Reference
IL-6	7.39 pg/mL	On admission	ARDS	[20]
Fibrinogen	5.16 g/L	On admission	Death	[113]
D-dimer	≥1 µg/mL	Outpatient	Death	[119]
LDH	445 µg/mL	On admission	Ventilation	[121]
CAC score	≥400	During hospitalization	Death	[123]
CRP	>40 mg/L	On admission	Death/ARDS	[124]
Ferritin	>950 ng/L	On admission	ARDS	[125]

ARDS: acute respiratory distress syndrome; CAC: coronary artery calcium; CRP: C-reactive protein; LDH: lactate dehydrogenase; IL-6: interleukin-6.

**Table 3 medicina-57-00833-t003:** Summary of key issues for older COVID-19 patients with CVD.

Issue	Mitigating Actions
Hypertension	Blood pressure control, do not discontinue ACEI or ARB medications
Diabetes	Early glycemic control
Risk of coagulopathy	Consider pharmacological thromboprophylaxis, especially if hospitalized
Access to healthcare	Virtual approach as necessary, family involvement if possible
Consequences of isolation: sedentary, poor diet, depression, anxiety, stress	Vaccination, nutritious diet, exercise program, social engagement (virtual or in person), mental and behavioral healthcare and monitoring

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
