# Peer review of "Cardiovascular Disease Complicating COVID-19 in the Elderly"

_medicina, 2021, doi:10.3390/medicina57080833_

Round 1
Reviewer 1 Report
overall a very good paper
The article titled Cardiovascular disease complicating COVID-19 in elderly provides an overview of cardiovascular complications in patients with COVID‐19. It identifies patients with sub-clinical atherosclerosis and are Covid positive may experience atherosclerotic plaque disruption and coronary thrombosis. They also report common symptoms related to hypertension and obesity resulting in heightened risk of mortality. Sedentary behaviors, poor diet and increased use of tobacco and alcohol associated with prolonged stay-at-home restrictions may promote thrombosis, while depressed mood due to social isolation can exacerbate poor self-care. The authors suggest telehealth interventions via smartphone applications and other technologies that document nutrition and offer exercise programs and social connection can be used to mitigate some of the potential damage to heart health.
This article does not appear to provide any new information related to CVD. The symptoms and complications stated are already known and existed prior to COVID. COVID just exacerbates what is already present. The strategies to address complications are already in use. No new information is being provided.
Author Response
Reviewer # 1 Comments
- COMMENT: This article does not appear to provide any new information related to CVD. The symptoms and complications stated are already known and existed prior to COVID. COVID just exacerbates what is already present. The strategies to address complications are already in use. No new information is being provided.
RESPONSE: We have updated this review to add new knowledge. Since it is a review, we do not claim to be adding original information. Rather, our goal is to put the current body of wisdom in a nicely organized and comprehensible form as a reference for healthcare professionals as well as others interested in the topic. Two new tables have been created and information on vaccines is now included.
We thank the reviewer and believe that the manuscript is improved as a result of their input. We hope you will agree, and decide in favor of accepting our report at this time.

Reviewer 2 Report
The manuscript titled "Cardiovascular Disease Complicating Covid-19 in the Elderly " is a very interesting article describing the CDV complications in patients with COVID-19. The manuscript is well written and references are updated and of good quality. However, authors should:
1) improve the introduction with a proper description of the CDV diseases in patients with cancer and the possible interactions with anticancer drugs and the relative risk of VTE ( you can cite doi: 10.3390/cancers12113316. )
2) describe the role of cytokines in COVID-19 -related CDV
3) describe, in discussion part of the manuscript, the potential lifestyle changes to suggest in these patients, for example a proper anti-inflammatory diet and/or nutraceuticals able to rduce the risk of myocarditis/heart failure of VTE in patients with COVID-19 ( you can cite doi: 10.26355/eurrev_202105_25957. )
The manuscript will be acceptable after minor revision.
Author Response
Reviewer # 2 Comments
- COMMENT: Improve the introduction with a proper description of the CDV diseases in patients with cancer and the possible interactions with anticancer drugs and the relative risk of VTE (you can cite doi: 10.3390/cancers12113316. )
RESPONSE: We have improved the introduction and cited the suggested paper (now reference 16) along with several others.
- COMMENT: Describe the role of cytokines in COVID-19 -related CDV
RESPONSE: A description of the role of cytokines in Covid-19 has been added on page 16 and appears throughout the text.
- COMMENT: Describe, in discussion part of the manuscript, the potential lifestyle changes to suggest in these patients, for example a proper anti-inflammatory diet and/or nutraceuticals able to reduce the risk of myocarditis/heart failure of VTE in patients with COVID-19 ( you can cite doi: 10.26355/eurrev_202105_25957. )
RESPONSE: We have added a description of the anti-inflammatory diet and nutraceuticals and cited the suggested paper (now reference 221) as well as several others.
We thank the reviewer and believe that the manuscript is improved as a result of their input. We hope you will agree, and decide in favor of accepting our report at this time.

Reviewer 3 Report
The authors reviewed the study “Cardiovascular Disease Complicating Covid-19 in the Elderly”. The cardiovascular disease in the Covid-19 remains the focus of the different experimental and clinical investigations and is well known as a major challenge in the Covid-19 disease and its related progression and long-term consequences. Based on these authors summarized the recent findings in this related topic indicated below, the authors could make some useful progress without a lot of additional investigations.
1. Comments
Introduction:
- The authors focused on older age and related hypertension. However, there are some wide discussions for such topics which might include other ages as well. Authors need to specify these studies and if possible, include a summarized table or response with these studies.
- The author's used the immune system as a cofactor with hypertension. However, this chapter might include inflammation and Covid-19. These studies include immune-related disorders in hypertension which might progress in a cytokine storm. However, authors need to specify that such studies are single specific to hypertension or other comorbidities as well. In addition, the authors specify tocilizumab and sarilumab as not encouraging drugs. Authors need to reconsider this once again due to sufficient evidence for their use and other immune suppressant drugs.
- The authors discussed the RAS in Covid-19 and related ACE inhibitors. Authors need to explain the relation of RAS disequilibrium in COVID-19 progression and following approaches in treatment concerning Mas Receptors, ACE-2 enzyme inhibition as well. In addition, authors need to discuss more continuation and discontinuation approaches that are nowadays which argue more the continuation rather than discontinuation. These are the latest recommendation from the Cardiovascular Societies and this speculation from the authors that ACE-2 receptor upregulation is not currently argued as well. We suggest the authors clarify this part more since there are many studies in the field which can support this discussion on the topic.
- There are also different pharmacogenetic studies in ACE enzymes that support the pathophysiology of COVID-19 in patients and the risk for the disease progression.
- The authors discussed Coagulopathies in COVID-19 and thromboprophylaxis. However, authors need to be incredibly careful in this setting since there are different approaches in the field which are still ongoing with major clinical trials regarding the dosage, class, and timeline of the use of anticoagulation therapy. In addition, the D-Dimer values are age-specific and also patients might include different previous Coagulopathies which are not related strictly to Covid-19.
- The authors used the Ferritin parameters in their table; however, they did not make the main connection of the increased Ferritin levels in Covid-19 and also the hypercoagulable state, hypoxia, etc. In addition, ferritin-targeted therapies are very promising in the early phases of the reduction of a hypercoagulable state.
- Authors end their study with effective vaccines and treatments. We suggest the author discuss the effects of prevention with vaccines and related outcomes in cardiovascular diseases as well in such populations.
- We suggest the authors include the summarized table and figure with their reviewing points to offer a more representative and general understanding.
Author Response
Reviewer # 3 Comments
- COMMENT : The authors focused on older age and related hypertension. However, there are some wide discussions for such topics which might include other ages as well. Authors need to specify these studies and if possible, include a summarized table or response with these studies.1.
RESPONSE: We have now composed a table with this information (Table 1).
- COMMENT: The author's used the immune system as a cofactor with hypertension. However, this chapter might include inflammation and Covid-19. These studies include immune-related disorders in hypertension which might progress in a cytokine storm. However, authors need to specify that such studies are single specific to hypertension or other comorbidities as well. In addition, the authors specify tocilizumab and sarilumab as not encouraging drugs. Authors need to reconsider this once again due to sufficient evidence for their use and other immune suppressant drugs.
RESPONSE: We have updated our discussion of tocilizumab and sarilumabon page 3 with new references 61 and 62.
- COMMENT: The authors discussed the RAS in Covid-19 and related ACE inhibitors. Authors need to explain the relation of RAS disequilibrium in COVID-19 progression and following approaches in treatment concerning Ras Receptors, ACE-2 enzyme inhibition as well. In addition, authors need to discuss more continuation and discontinuation approaches that are nowadays which argue more the continuation rather than discontinuation. These are the latest recommendation from the Cardiovascular Societies and this speculation from the authors that ACE-2 receptor upregulation is not currently argued as well. We suggest the authors clarify this part more since there are many studies in the field which can support this discussion on the topic.
RESPONSE: We have updated our discussion of ACEI and ARBs in COVID-19 patients and have made it clear that the current recommendation is to continue these drugs in those already taking them as the benefits outweigh the risks in the setting of COVID-19. This information is now on pages 5-6 and again on page 12 with new references 84-87, 91, 245, 246.
- COMMENT: There are also different pharmacogenetic studies in ACE enzymes that support the pathophysiology of COVID-19 in patients and the risk for the disease progression.
RESPONSE: We have recognized this in our discussion on page 6 and added appropriate references 89 and 90 as well.
- COMMENT: The authors discussed Coagulopathies in COVID-19 and thromboprophylaxis. However, authors need to be incredibly careful in this setting since there are different approaches in the field which are still ongoing with major clinical trials regarding the dosage, class, and timeline of the use of anticoagulation therapy. In addition, the D-Dimer values are age-specific and also patients might include different previous Coagulopathies which are not related strictly to Covid-19.
RESPONSE: We have added information about the age-specific nature of D-dimer values and other coagulopathies. This can be found on page 7 with new reference 129.
- COMMENT: The authors used the Ferritin parameters in their table; however, they did not make the main connection of the increased Ferritin levels in Covid-19 and also the hypercoagulable state, hypoxia, etc. In addition, ferritin-targeted therapies are very promising in the early phases of the reduction of a hypercoagulable state.
RESPONSE: We apologize for not making these connections clear and have now done so in the manuscript. Ferritin-targeted therapies are included in the revision on pages 9-10 with new references 177-180.
- COMMENT: Authors end their study with effective vaccines and treatments. We suggest the author discuss the effects of prevention with vaccines and related outcomes in cardiovascular diseases as well in such populations.
RESPONSE: We have added this discussion of vaccines to the section on Tailoring Treatment and changed the section title to: “Tailoring CVD Treatment and Vaccination in the Context of COVID-19” on pages 11-12
- COMMENT: We suggest the authors include the summarized table and figure with their reviewing points to offer a more representative and general understanding.
RESPONSE: We have added this Table (Table 3).
We thank the reviewer and believe that the manuscript is improved as a result of their input. We hope you will agree, and decide in favor of accepting our report at this time.
